# New Functionalized Chitosan with Thio-Thiadiazole Derivative with Enhanced Inhibition of Pathogenic Bacteria, Plant Threatening Fungi, and Improvement of Seed Germination

Ahmed G. Ibrahim [1,*], Walid E. Elgammal [1], Ahmed M. Eid [2], Maha Alharbi [3], Ahmad E. Mohamed [4], Aisha A. M. Alayafi [5], Saber M. Hassan [1] and Amr Fouda [2,*]

1   Department of Chemistry, Faculty of Science, Al-Azhar University, Nasr City, Cairo 11884, Egypt; walidebaied.sci85@azhar.edu.eg (W.E.E.)
2   Department of Botany and Microbiology, Faculty of Science, Al-Azhar University, Nasr City, Cairo 11884, Egypt; aeidmicrobiology@azhar.edu.eg
3   Department of Biology, College of Science, Princess Nourah bint Abdulrahman University, P.O. Box 84428, Riyadh 11671, Saudi Arabia; maalharbi@pnu.edu.sa
4   Delta Aromatic International Company, First Industrial Zone, 6th of October, Giza 12566, Egypt
5   Biological Sciences Department, Faculty of Science, University of Jeddah, Jeddah 21577, Saudi Arabia; aamalayafy@uj.edu.sa
*   Correspondence: ahmed_polytech@azhar.edu.eg (A.G.I.); amr_fh83@azhar.edu.eg (A.F.)

**Abstract:** In this study, a new modified chitosan conjugate (Chito-TZ) was developed via amide coupling between the acid chloride derivative of the methylthio-thidiazole compound and the free primary amino groups of chitosan. The product was characterized using several instrumental investigations, including Fourier-transform infrared spectroscopy (FT-IR), $^1$H-Nuclear magnetic resonance, X-ray photoelectron spectroscopy (XPS), thermogravimetric analysis (TGA), and X-ray diffraction (XRD). XRD indicated that the crystalline pattern of chitosan was interrupted after chemical modification with the thiadiazole derivative. Broido's model was used to determine the thermal activation energy $E_a$, and the results showed that the $E_a$ for the first decomposition region of Chito-TZ is 24.70 KJ mol$^{-1}$ lower than that required for chitosan (95.57 KJ mol$^{-1}$), indicating the accelerating effect of the thiadiazole derivative on the thermal decomposition of Chito-TZ. The modified chitosan showed better antibacterial and antifungal activities than the non-modified chitosan; except for seed germination, chitosan was better. The Chito-TZ showed a low MIC value (25–50 μg mL$^{-1}$) compared to Chito (50–100 μg mL$^{-1}$). Moreover, the maximum inhibition percentages for plant-pathogenic fungi, *Aspergillus niger*, *Fusarium oxysporum*, and *Fusarium solani*, were attained at a concentration of 300 μg mL$^{-1}$ with values of 35.4 ± 0.9–39.4 ± 1.7% for Chito and 45.2 ± 1.6–52.1 ± 1.3% for Chito-TZ. The highest germination percentages (%) of broad bean, shoot and root length and weight, and seed vigor index were obtained after Chito treatment with a concentration of 200 μg mL$^{-1}$ compared to Chito-TZ.

**Keywords:** chitosan conjugates; thiadiazole derivative; Broido's model; pathogenic fungi; seed germination

## 1. Introduction

There is great research interest in chitosan because of its unique properties, which are non-toxicity, biocompatibility, biodegradability, and antibacterial and antioxidant activity, in addition to being a derivative of a natural polymer, chitin, which is considered the second natural polymer after cellulose [1,2]. These properties make it and its derivatives prevalent in many applications, such as adsorbents for pollutants in wastewater treatment, packaging materials, cosmetics, antioxidant, and antibacterial agents, as well as in the biomedical healthcare fields, including wound healing, scaffolds for tissue engineering, drug delivery systems, and antitumor chemotherapy [3–6].

Chitosan chains are composed of randomly distributed repeating units of glucosamine and acetylglucosamine in ratios that depend on the reaction conditions for alkaline deacetylation of chitin, and chitosan is the only polysaccharide that contains free amino groups, which plays an important role not only in its biological activity and physicochemical properties but also in its possibility to be modified chemically [7]. The biological activity and physicochemical properties of chitosan are categorized based on the number of free amino groups (degree of deacetylation, DD) as well as the molecular weight. Chitin is leached from the exoskeleton of crustaceans (shrimp, lobster, and crab shells), arthropods, fungal cell walls, and insects [8]. Chitosan is considered a cationic polymer, so it can form complexes via electrostatic interaction with negatively charged particles like anionic polymers and anionic pollutants, as well as the anionic groups on the surface of bacteria [9–11].

Although native chitosan is used in some applications, such as water treatment, packaging materials, and antimicrobial agents, its poor solubility in common organic solvents and aqueous solutions hinders its spread in many applications [12]. Chemical modification of chitosan chains, from the attachment of functional units to primary amino groups or hydroxyl groups, may contribute to improving its solubility and increasing its use in many industrial and pharmaceutical applications [13]. Several researchers transformed the primary amino groups into imine (–N=CH–) and amide (–NH–CO–) groups via chemical reactions. In [14], the synthesis of 1,2,3-triazole nicotinate chitosan conjugate on the primary amino group was described, and the in vitro antifungal activity against three crop-threatening fungi (*R. solani*, *S. solani*, and *A. porri*) was evaluated. In [15], the chitosan was modified via the Schiff base reaction on the amino group using two indole-3-carbaldehyde derivatives and evaluated against *S. aureus*, *E. coli*, and *C. albicans*, studying the antioxidant and in vitro cytotoxicity properties. Huang et al. [16] conducted a vanillin-chitosan Schiff base reaction, converting amino groups into imine groups for sustained release of zinc supplements. The researchers in [17] synthesized and characterized chitosan-g-amino anthracene derivatives, which showed fluorescence properties and enhanced inhibition against *E. coli*. Aytekin et al. [18] synthesized and evaluated the antioxidant activities of the chitosan-caffeic acid conjugates. The authors in [19] synthesized salicylaldehyde-chitosan Schiff base adsorbents for anionic dyes, and they found that the modification improved the adsorption efficiency better than chitosan. By Tamer et al. [20], the antimicrobial property of chitosan was improved against *E. coli*, *P. aeruginosa*, *Salmonella* sp., *S. aureus*, *B. cereus*, and *C. albicans* via chitosan conjugation with two aromatic aldehydes (4-chlorobenzaldehyde and benzophenone). Although there are some works [1,21–23] that report the biological activity of chitosan-thiadiazole conjugates, the inhibition effects of these conjugates against plant-threatening fungi and seed germination were still unstudied as per our survey in the reliable sources.

Here, our aim is to design a new antibacterial and non-toxic chitosan conjugate by altering its structure with a thio-thidiazole-based compound. In this study, we conduct the modification reaction via amide linkage between the primary amino groups of chitosan chains and the acid chloride derivative of the thio-thidiazole compound. The new chitosan conjugate (coded here as Chito-TZ) was investigated using several instrumental analyses, including FT-IR, [1]H-NMR, elemental analysis, TGA, SEM, XRD, and XPS. The antimicrobial activity against various pathogens (Gram-negative bacteria (*Escherichia coli* and *Pseudomonas aeruginosa*); Gram-positive bacteria (*Bacillus subtilis* and *Staphylococcus aureus*); unicellular fungus (*Candida albicans*); and the antifungal activity against three crop-threatening fungi (*Aspergillus niger*, *Fusarium oxysporum*, and *Fusarium solani*) were discussed via in vitro measurements. Moreover, the efficacy of Chito and Chito-TZ to enhance the broad bean seed germination was also investigated.

## 2. Materials and Methods

### 2.1. Materials

Chitosan (DD, 70–95%) and iodomethane (purity: ≥99.0%) were obtained from Sigma–Aldrich, (Cairo, Egypt). Thiosemicarbazide (purity: 98.0%), carbon disulfide (Purity:

99.8%), thionyl chloride (purity: 99.0%), and triethylamine (purity: 98.5%) were purchased from Loba Chemie, (Cuffe Parade, Mumbai, India). Solvents and other reagents such as dichloromethane (DCM, purity: 99.0%), dimethyl sulfoxide (purity: 99%), ethanol (purity: 99%), potassium hydroxide (purity: 99.0%), hydrochloric acid (purity: 37.0%), and anhydrous sodium carbonate (purity: 99.0%) were bought from Fisher chemicals, Thermo Fisher Scientific Inc., (Waltham, MA, USA) and El-Nasr companies, (Cairo, Egypt) respectively. Distilled water was prepared in our research lab.

### 2.2. Methods

#### 2.2.1. Synthesis of 5-Amino-1,3,4-thiadiazole-2-thiol (Compound **1**)

The title compound was prepared according to the procedure that matched the previously reported method to give the product [1,24] (yellowish-white crystals, yield: 78%, melting point: 238–239 °C).

#### 2.2.2. Synthesis of 5-(Methylthio)-1,3,4-thiadiazol-2-amine (Compound **2**)

The title compound was prepared using the following method: To a cold solution of ethanolic potassium hydroxide (0.1 mol/20 mL) in a 100-mL Erlenmeyer conical flask equipped with a magnetic stirring bar, 5-amino-1,3,4-thiadiazole-2-thiol (**1**) (0.1 mol) was added in small lots. After 10 min of stirring the reaction, the appropriately weighed methyl iodide was added for a period of 15 min, and then the cooling bath was removed. The crude reaction mixture was stirred for 8 h at room temperature (rt). Upon completion, the precipitate formed was collected via filtration, washed with water, and then recrystallized from ethanol/water to give compound **2** (yellowish-white crystal, yield: 91%, melting point: 128–130 °C).

#### 2.2.3. Synthesis of 4-((5-(Methylthio)-1,3,4-thiadiazol-2-yl)amino)-4-oxobutanoic Acid (Compound **3**)

To a solution of compound **2** (0.01 mol) in benzene (25 mL), succinic anhydride (0.01 mol) was added, and the resulting reaction mixture was vigorously stirred for 3 h at room temperature. The resulting precipitate was filtered and washed with ethanol and air-dried, followed by crystallization from the ethanol and benzene mixture, resulting in an off-white solid with a yield of 81%, a melting point of 262–264 °C, and elemental analyses (%) of C, 34.00; H, 3.67; N, 16.99; and S, 25.93. Found (%): C, 33.89; H, 3.58; N, 16.88; S, 25.82.

#### 2.2.4. Synthesis of Chitosan Derivative (Chito-TZ)

The modified chitosan was prepared as follows: An acyclic chlorinated agent, including thionyl chloride (0.013 mol), was added portion-wise while maintaining the reaction temperature between 0 and 10 °C to a solution of 1,3,4-thiadiazol carboxylic acid derivative **3** (0.01 mol) in anhydrous DCM (50 mL). After the addition was finished, the reaction mixture was heated for two hours at a refluxing temperature of 70 °C. In order to provide a crude product for the subsequent step without purification, the reaction mixture was then evaporated as a result of the reduced pressure. Chitosan (0.01 mol) was added to the crude product in 30 mL of DCM ($CH_2Cl_2$) in the presence of triethylamine (0.03 mol). The reaction mixture was stirred at 70 °C overnight. The mixture is cooled, filtered via a Büchner funnel, and rinsed with dichloromethane to afford the new modified chitosan derivative, as portrayed in Scheme 1.

### 2.3. Instrument Specifications

In this study, several instrumental investigations like elemental analysis, Fourier-transform infrared spectroscopy (FT-IR), $^1$H/$^{13}$C nuclear magnetic resonance, mass spectrometry, X-ray photoelectron spectroscopy (XPS), scanning electron microscopy (SEM), thermogravimetric analysis (TGA), and X-ray diffraction (XRD) were utilized to confirm the synthesis products. Descriptions of the utilized instruments are presented in Table 1.

**Scheme 1.** Schematic procedure for the synthesis of Chito-TZ derivative.

**Table 1.** Descriptions of the utilized instruments in this study.

| Instrument | Description |
|---|---|
| Elemental analyzer | AnalyzerVario El M, Germany, Elements C-H-N-S, |
| FT-IR | Schimadzu, Nicolet iS10FT IR (Thermo Fisher Scientific, Waltham, MA, USA), resolution 16, scanning range 4000–400 cm$^{-1}$, KBr plates, wavenumber scale (cm$^{-1}$). |
| $^1$H/$^{13}$C-NMR | JNM-ECA 500 II/JEOL-JAPAN instrument, solvent DMSO-$d_6$, at 500 MHz for $^1$H and 125 MHz for $^{13}$C), δ scale (ppm) downfield from TMS. |
| Mass analysis | Shimadzu Japan's GC–2010 |
| SEM | SEM (FEI inspect 5 with an EDX unit, Holland) with an operating voltage of 20–30 KV. |
| XPS | K-ALPHA (Thermo Fisher Scientific, Waltham, MA, USA), monochromatic x-ray Al K-Alpha radiation from −10 to 1350 e.v. Spot size 400 micro m at pressure 10–9 mbar with full spectrum pass energy 200 e.v. and at narrow spectrum 50 e.v. |
| TGA | Discovery SDT 650-Simultaneous DSC-TGA/DTA Instruments, New Castle, DE, USA, N$_2$ atmosphere, temperature range 0 to 500 °C, heating rate 10 °C/min. |
| XRD | X'Pert powder PAN analytical, Cu-radiation (l = 1.542 Å), 45 K.V., 35 M.A. scanning speed 0.02°/sec. |

*2.4. Biological Analysis*

2.4.1. Antimicrobial Activity

To investigate the antimicrobial properties of modified chitosan (Chito-TZ) compared to unmodified chitosan (Chito) against various pathogens, including *Escherichia coli* ATCC8739, *Pseudomonas aeruginosa* ATCC9022 (Gram-negative bacteria), *Bacillus subtilis* ATCC6633, *Staphylococcus aureus* ATCC6538 (Gram-positive bacteria), and *Candida albicans* ATCC10231 (unicellular fungus), the agar-well diffusion assay was used [25]. The *C. albicans* and bacterial strains were cultured in yeast extract peptone dextrose (YEPD) and nutrient broth media, respectively, for 24 h at 35 ± 2 °C. Muller Hinton agar media (Oxoid) was sterilized and poured into sterilized Petri dishes under aseptic conditions. After solidification, each strain was uniformly inoculated over the plate surface using a sterilized swab. Four wells were made in each solidified inoculated plate using cork borer (0.7 mm). Different concentrations (300, 200, 100, 50, 25, and 12.5 µg mL$^{-1}$) of Chito-TZ and Chito were prepared in DMSO (*w/v*). Then, 100 µL of each concentration, as well as pure DMSO as a negative control, were poured into an agar well. The laden Muller–Hinton agar plates were refrigerated for 60 min and then incubated at 35 ± 2 °C for 24 h to determine the activity by measuring the zone of inhibition (ZOI). The lowest concentration that caused an inhibition zone was determined as the minimum inhibitory concentration (MIC) [26]. The experiment was performed with three independent repetitions.

2.4.2. Activity of Chito-TZ and Chito against Phytopathogen

The antifungal activity of both Chito and Chito-TZ was evaluated against several phytopathogenic fungi designated as *Aspergillus niger*, *Fusarium solani*, and *Fusarium oxysporum*, obtained from the Desert Research Center in El-Mataria, Cairo, Egypt. In this assay, an agar plug (8 mm) of each fungal strain, covered with heavy-growing mycelia, was placed in the center of potato dextrose agar (PDA) media, which was supplemented with 200 µL of modified or non-modified chitosan prepared at various concentrations (300, 200, and 100 µg mL$^{-1}$). The inoculated plates were then incubated at 28 ± 2 °C for six days. After incubation, the radial growth of each fungal strain was measured, and the growth inhibition was calculated using the following Equation (1) [27]:

$$\text{Growth inhibition percentages (\%)} = \frac{\text{Radial growth of control} - \text{Radial growth of treated sample}}{\text{Radial growth of control}} \times 100\% \quad (1)$$

where radial growth of control is the diameter of fungal growth after six days incubated under the same incubation conditions in absence of treatment.

2.4.3. Seed Germination

To evaluate the effect of Chito and its derivative, Chito-TZ, on seedling growth, broad bean (*Vicia faba*) seeds were used following a standard method [28]. Healthy seeds were obtained from the local market and surface sterilized by immersing them in a 10% sodium hypochlorite solution for 10 min, followed by thorough washing with sterilized dH$_2$O. Sterilized seeds were then placed in sterile Petri plates (90 × 15 mm) lined with filter paper wetted with 5.0 mL of different concentrations (100, 200, and 300 µg mL$^{-1}$) of Chito-TZ or Chito. Seeds incubated with sterile dH$_2$O were used as control. Each treatment was performed in triplicate with 10 seeds on each plate. The Petri plates were incubated at 28 ± 2 °C in a dark condition for 10 days, and data were recorded for germination percentages (%), shoot length, root length, shoot weight, and root weight. Also, the seed vigor index (SVI) was calculated using the following Equation (2):

$$\text{SVI} = \text{germination percentages} \times \text{shoot seedling length} \quad (2)$$

2.4.4. Statistical Analysis

The biological activity data were obtained from three independent replicates and analyzed using SPSS v17 statistical software. The mean differences among the treatments

were evaluated using either the *t*-test or analysis of variance (ANOVA), followed by the Tukey HSD test with a significance level of $p < 0.05$.

## 3. Results and Discussion

### 3.1. Synthesis of the Chitosan Conjugate

This paper developed a novel chitosan conjugate with a 1,3,4-thiadiazole group using the method epitomized in Scheme 1. At first, the 1,3,4-thiadiazole derivative **1** was synthesized by cyclizing thiosemicarbazide with carbon disulfide in the presence of an anhydrous inorganic catalyst, such as anhydrous sodium carbonate ($Na_2CO_3$), under refluxing temperature for 6 h in pure ethanol, which was alkylated in ethanolic potassium hydroxide (alc. KOH) at room temperature with iodomethane ($CH_3I$) for 4 h to give the S-methyl derivative **2**, as portrayed in Scheme 1. Furthermore, acid derivative **3** was prepared by stirring compound **2** with succinic anhydride ($C_4H_4O_3$) at room temperature for 3 h in benzene as a solvent, which was subjected to a chlorination reaction with thionyl chloride ($SOCl_2$) to give the non-isolable intermediate (acid chloride), which was introduced into the skeleton of chitosan via coupling with the free primary amino groups of chitosan. Various spectral data were used to evaluate the newly synthesized compounds and chitosan conjugate, and each of the spectral results was compatible with the predicted structures.

### 3.2. Spectral Analysis

FT-IR (KBr, $\upsilon_{max}/cm^{-1}$) of compound **2** (Figure 1a) exhibited absorption peaks, in accordance with [24,29], at 3275 $cm^{-1}$ and 3108 $cm^{-1}$ belonging to the stretching vibrations of the $NH_2$ group, while the peak at 2948 $cm^{-1}$ was the symmetric stretching vibration of aliphatic C–H in $CH_3$. Also, the absorption peaks at 1629 $cm^{-1}$ and 1521 $cm^{-1}$ were correlated with the stretching vibrations of C=N and C=C groups, respectively. The peak at 1431 $cm^{-1}$ appears to be due to the C–H bending vibration. On the other hand, the FT-IR (KBr, $\upsilon_{max}/cm^{-1}$) (Figure 1b) of derivative **3** showed characteristic peaks at 3268 $cm^{-1}$, 3170 $cm^{-1}$, 2928 $cm^{-1}$ and 2853 $cm^{-1}$, which corresponded to the stretching vibrations of O-H, N-H, and aliphatic C-H (symmetric and asymmetric) [30]. The peaks of the stretching vibrations for 2C=O, C=N, and C=C were found at 1686 $cm^{-1}$, 1582 $cm^{-1}$, and 1440 $cm^{-1}$, respectively. While the peak at 1332 $cm^{-1}$ corresponds to O–H bending vibrations. In addition, the FT-IR spectrum (KBr, $\upsilon_{max}/cm^{-1}$) of Chito is displayed in Figure 1c. All observed peaks were in line with those reported in previous articles [31,32]. Broad peaks at 3331 $cm^{-1}$ and 3291 $cm^{-1}$ are related to the stretching vibrations of the O-H and $NH_2$ groups, and the peaks at 2921 $cm^{-1}$ and 2877 $cm^{-1}$ are ascribed to the stretching vibrations of aliphatic C–H (symmetric and asymmetric). The intensity of the peak at 1645 $cm^{-1}$ is assigned to C=O stretching ((amide I) residual N-acetyl groups) [33]. The bending vibrations of the N–H group appeared at 1589 $cm^{-1}$. Two weak peaks at 1423 $cm^{-1}$ and 1375 $cm^{-1}$ were due to the bending vibrations O-H in $CH_2OH$ and the $CH_2$ in $CH_2OH$ (wagging and twisting), and the peaks at 1262 $cm^{-1}$, 1325 $cm^{-1}$, and 1154 $cm^{-1}$ attributed to the symmetrical stretching vibrations of the $CH_3$ in $N-COCH_3$, the stretching vibrations of the C-N (amide III), and the stretching vibrations of C-O in $CH_2OH$, whereas the other peaks manifested at 1066 $cm^{-1}$ and 1028 $cm^{-1}$, corresponding to the C-O-C bridge and C-O (stretching vibrations for both symmetric and asymmetric), and 896 $cm^{-1}$ associated with the pyranose ring. Meanwhile, the chemical composition of the chitosan conjugate Chito-TZ was analyzed using (KBr, $\upsilon_{max}/cm^{-1}$) (Figure 1d). The amide bond formation between the chitosan and 1,3,4-thiadiazol carboxylic acid derivatives was demonstrated by a shift of the FT-IR wavenumber of 1,3,4-thiadiazol carboxylic acid from 1686 $cm^{-1}$ to 1693 $cm^{-1}$, and the occurrence of the peak at 3191 $cm^{-1}$ was related to the stretching vibrations of N-H. Moreover, obvious changes can be seen in the shift of the peak of the OH group from 3331 $cm^{-1}$ to 3417 $cm^{-1}$, and more subtle changes appear between 1338 $cm^{-1}$ and 428 $cm^{-1}$. From the above, we conclude that the spectral data reveals that the new chitosan conjugate has been achieved.

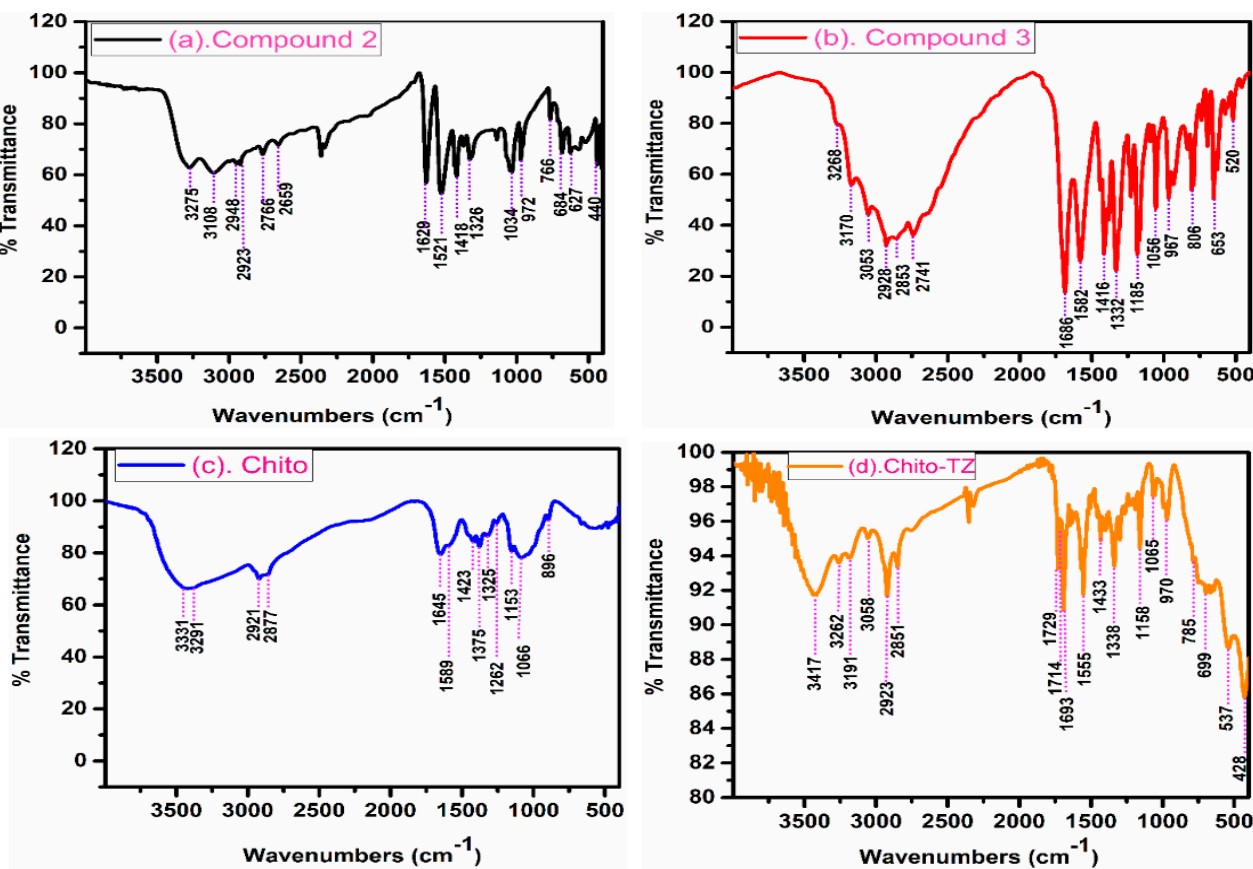

**Figure 1.** FT-IR spectra of (**a**) compound **2**, (**b**) compound **3**, (**c**) Chito, and (**d**) Chito-TZ.

The NMR spectra of compound **3** are revealed in Figure 2. In Figure 2a, the $^1$H NMR signals at 0 ppm, 2.5 ppm, and 3.3 ppm correspond to the signals of tetramethylsilane (TMS), residual non-deuterated dimethyl sulfoxide (DMSO), and residual $H_2O$ in the sample, respectively. The characteristic singlet signals at 12.56 ppm and 12.22 ppm (exchangeable with $D_2O$-DMSO in Figure 2b) were attributed to -OH and amidic -NH protons, respectively. Also, a singlet signal at 2.65 ppm referred to the three protons of the methyl thiol ($SCH_3$) group and showed triplet signals at 2.52 ppm and 2.46 ppm representing the protons of the methylene groups ($-CH_2-CH_2-$). On the other hand, the $^{13}$C NMR analysis (126MHz, DMSO-$d_6$, Figure 2c) revealed the carbon of the carbonyl groups of (-COOH) and -NH-CO- at 174.01 ppm and 171.21 ppm, respectively. The thiadiazole ring's carbons were detected at 160.60 ppm and 158.69 ppm. Additionally, the signals at 30.33 ppm, 28.82 ppm, and 16.44 ppm were due to the carbon in $CH_2$ (acidic), $CH_2$ (amidic), and methyl thiol ($SCH_3$), respectively. Furthermore, the structure of compound 3 was also confirmed by the mass spectrum (Figure 2d), which was consistent with the proposed structure. The suggested structure involves the successive losses of $-H_2O$, $-CH_2O_2$, $-C_2H_2$, -CO, -H, -SH,-$CH_2$,-$SCH_2$, -CN, -HCN, and $-C_3H_4N_3S_2$ resulting in ion fragment peaks at 229 (53.57%), 201 (2.33%), 175 (1.8%), 146 (5.69%), 142 (1.73%), 133 (0.41%), 101(20.03%), 100 (6.95%), 74 (2.21%), and 55 (56.39%), respectively, and the most prominent peak (base peak) was seen at 147 (100%), which was attributed to 5-(methylthio)-1,3,4-thiadiazol-2-amine ($C_3H_5N_3S_2$) fragment ion.

As reported previously in [34,35], the $^1$H-NMR spectrum of **Chito** showed three protons of N-acetyl glucosamine at chemical shifts of 1.93 ppm, and the signal at 3.18 ppm was ascribed to the H-2 proton of glucosamine residues. Also, the signals emanating from 3.25 to 4.00 ppm were attributed to the non-anomeric protons (H-3, 4, 5, and 6) and a signal of anomeric protons (H-1) of glucosamine residues was found at 4.66 ppm. In Figure 3, the $^1$H-NMR (500 MHz, DMSO-$d_6$) of Chito-TZ, in contrast to CS, recorded a chemical shift of 2.66 ppm and 2.46 ppm, which were attributed to the protons of ($SCH_3$) and two methylene

units from the succinyl group, respectively. Furthermore, the singlet signal at 12.61 ppm was the chemical shift of N-H for (NH-CO). Also, as seen in Figure 3, it was demonstrated that 1,3,4-thiadiazol carboxylic acid was successfully grafted onto free amino groups of chitosan via the acylation reaction, as described by the appearance of a new singlet signal at 7.95 ppm, which was attributed to the resonance of the (N-H) proton, and the absence of the characteristic signal, which was related to the resonance of the hydroxyl unit at 12.56 ppm.

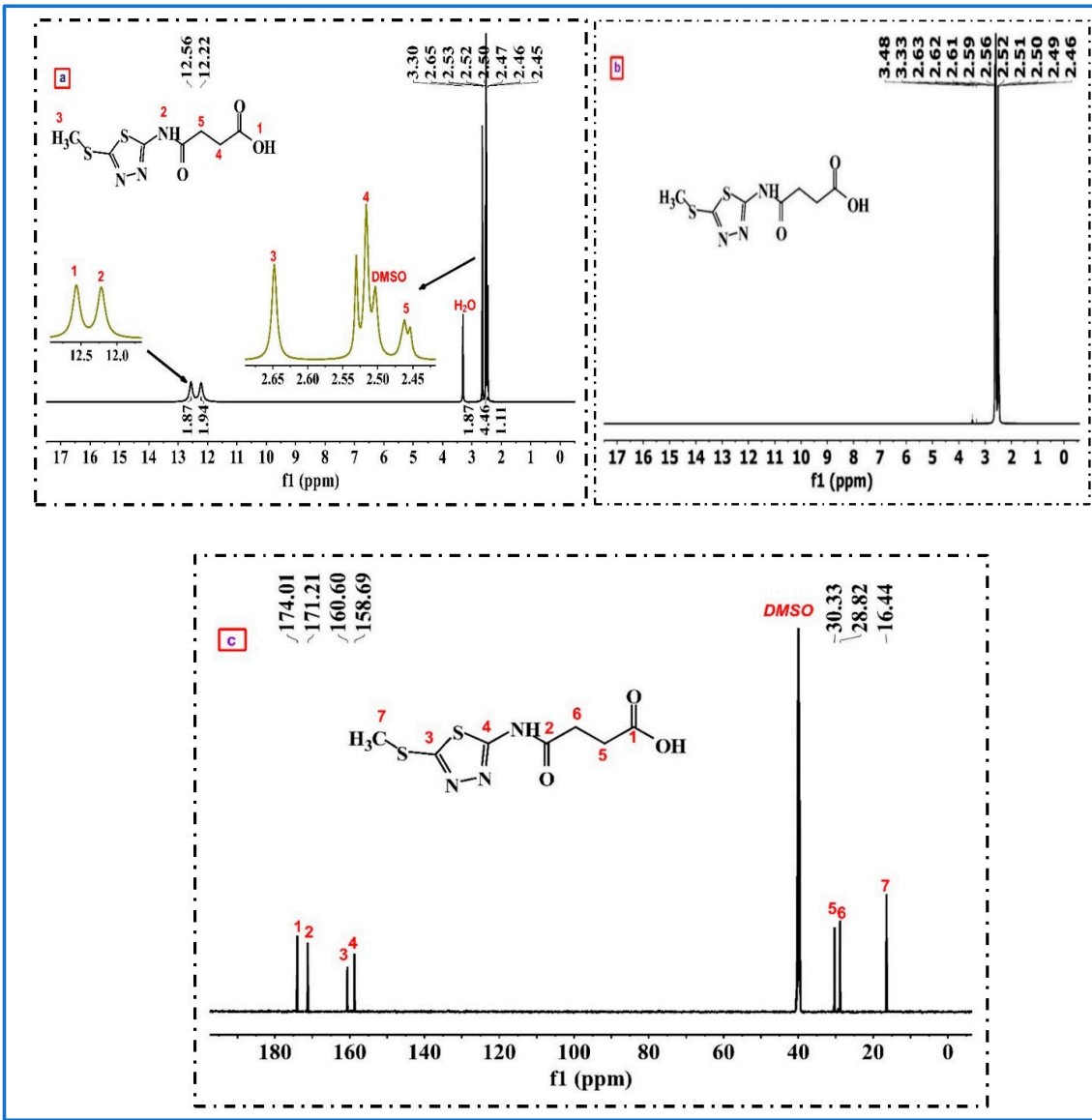

**Figure 2.** *Cont*.

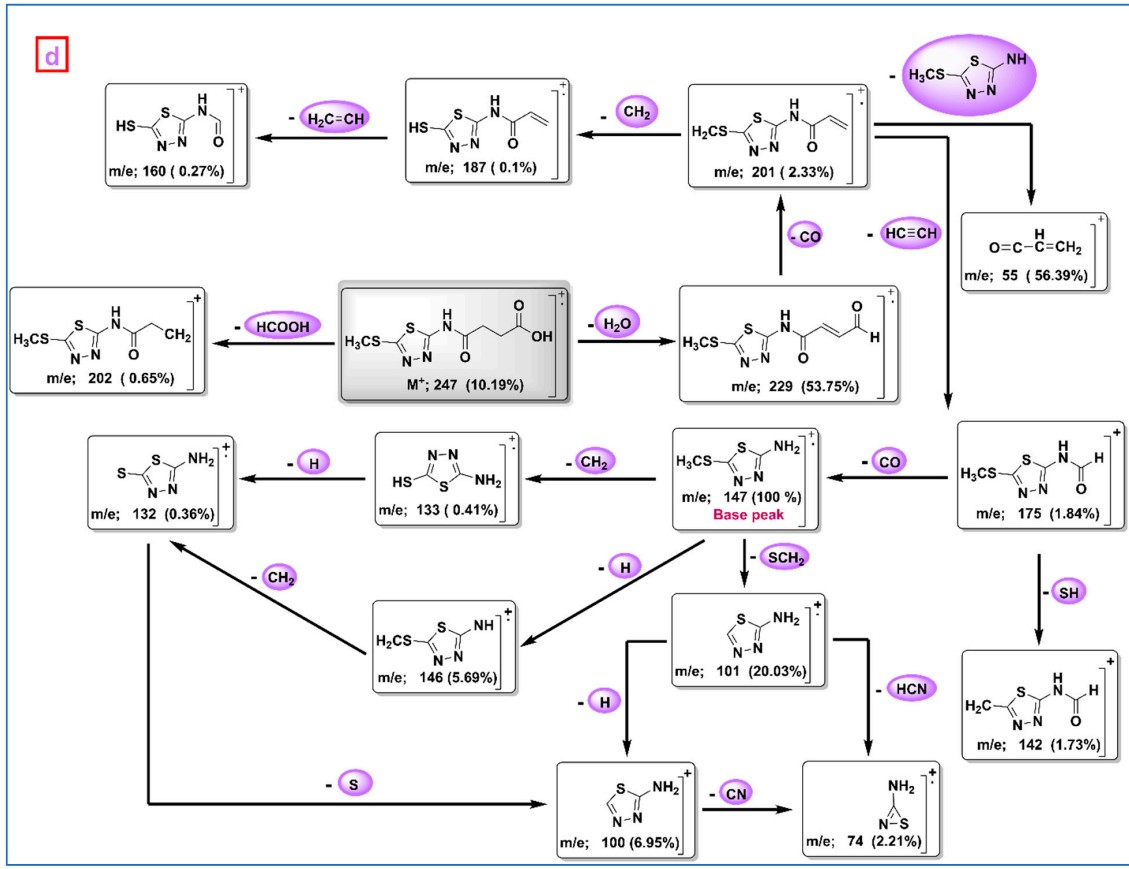

**Figure 2.** NMR spectra of compound 3 (**a**) ¹H-NMR spectrum, (**b**) ¹H-NMR-D2O spectrum, and (**c**) ¹³C-NMR spectrum; (**d**) suggested fragmentation pattern of compound 3.

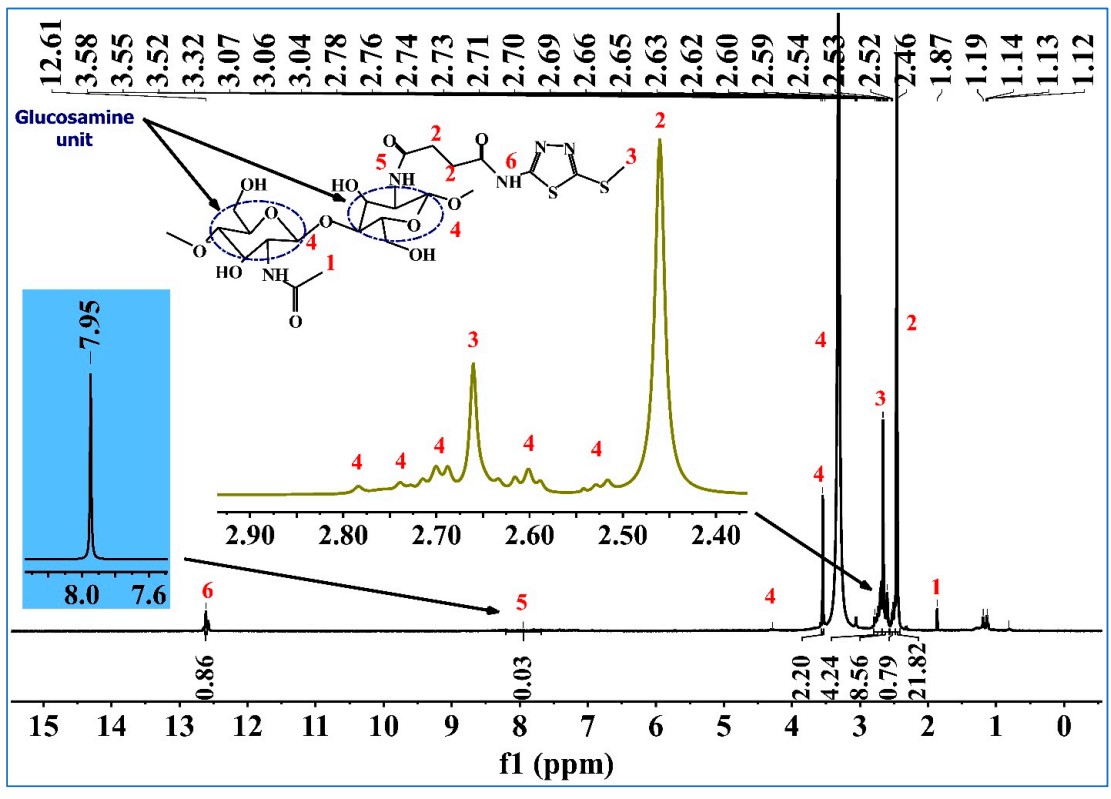

**Figure 3.** ¹H-NMR spectrum of Chito-TZ.

### 3.3. Elemental Analysis

The structure difference between Chito and Chito-TZ was indicated by the analysis of C, H, N, and S elements. While the Chito showed elemental analysis (%) 36.38, 5.56, and 6.05 for C, H, and N, respectively, the Chito-TZ showed 27.85, 4.22, and 6.88, respectively, in addition to 13.21% for the S element. The appearance of the S element in the elemental analysis of Chito-TZ is evidence of the modification reaction. The ratio (C/N) was used for determining the substitution degree from the following Equation (3) [36]:

$$\text{Substitution degree} = \frac{X(C/N)_a - (C/N)_b}{Y} \tag{3}$$

where a is the (C/N) ratio for Chito-TZ, b is the (C/N) ratio for Chito, and X and Y are $N_{Chito-TZ}$-$N_{Chito}$ and $C_{Chito-TZ}$-$C_{Chito}$, respectively. The substitution degree was found to be 0.96.

### 3.4. SEM Investigation

SEM images of Chito and Chito-TZ are presented in Figure 4. It was clearly observed that the surface morphology of chitosan changed after the modification reaction with the thiadiazole derivative. The surface of chitosan was smooth with minor wrinkles, but in contrast, Chito-TZ appeared with a different surface with high roughness. This change in chitosan surface was in accordance with previous reports [16,37] and was a result of the chemical modification reaction with methylthio-thiadiazolyl amino-4-oxobutanoic acid.

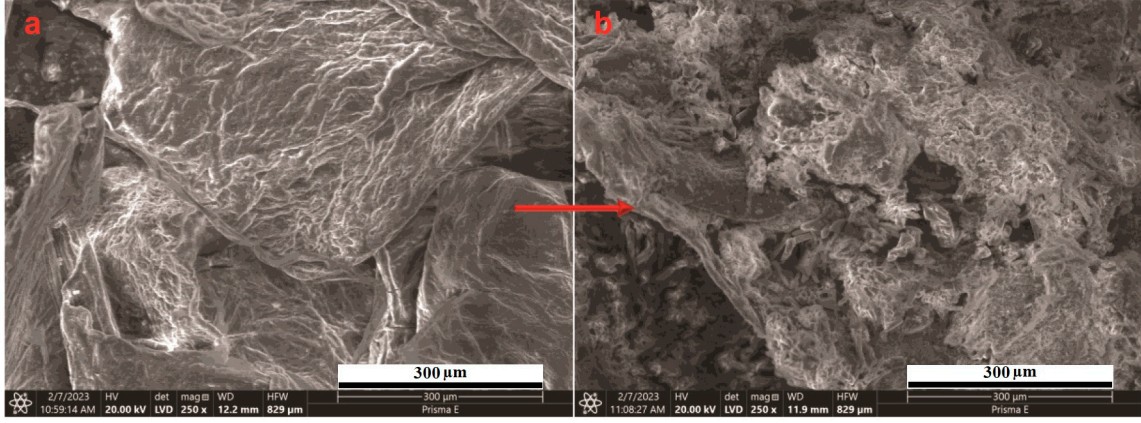

**Figure 4.** SEM images of (**a**) Chito and (**b**) Chito-TZ.

### 3.5. Thermal Properties

Figure 5 shows the TGA analysis of Chito and Chito-TZ in a non-oxidized (nitrogen) atmosphere. As shown in Figure 5, the chitosan decomposes in one stage, whereas the chitosan conjugate, Chito-TZ, decomposes in three stages. This is not the only difference between Chito and Chito-TZ. The $T_{10}$ (temperature of 10% weight loss) and $T_{20}$ (temperature of 20% weight loss) are also different. These differences confirm the success of the modification reaction of chitosan. Chitosan showed a weight loss of 9.53% at 123.2 °C, which is attributed to the evaporation of weakly bonded and/or physically adsorbed water from chitosan, whereas Chito-TZ showed only 6.82% at the same temperature because of the reduced amount of free amino groups that adsorb the water molecules through the hydrogen bonding. The main chains of chitosan decompose at temperatures ranging from 266.04 to 310.16 °C via depolymerization reactions as well as the formation of volatile compounds. The thiadiazole derivative in Chito-TZ starts to decompose initially at 141.9 °C and continues to 233.2 °C, causing a weight loss of 14.69%. In the same temperature range, chitosan showed only 1.90% weight loss. The second decomposition stage of Chito-TZ occurs in the temperature range of 233.2–245.3 °C due to the initial cleavage of the main glucoside chain. The third stage within the range of 245.3–332.6 °C corresponds to the

continuous decomposition reactions such as depolymerization, dehydration, deamination, the opening of the glucopyranose rings, and the formation of volatile compounds. From Table 2, the values of $T_{10}$ for Chito and Chito-TZ were 133.8 °C and 178.5 °C, and the difference between $T_{10}$ and $T_{20}$ for Chito was 142.7 °C, while it was 47.7 °C for Chito-TZ, indicating the accelerating effect of the thiadiazole derivative on the decomposition reactions of Chito-TZ. Based on these interpretations, we can conclude that the Chito-TZ structure is less thermally stable than chitosan.

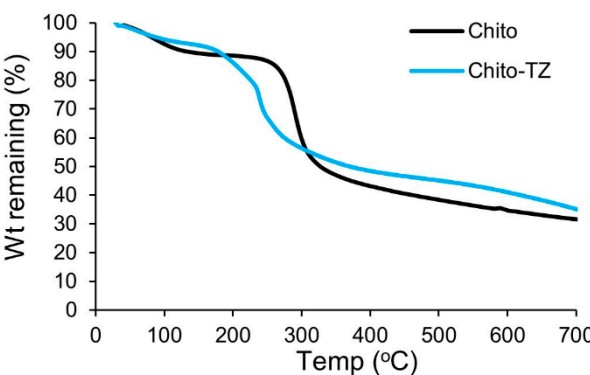

**Figure 5.** Thermogravimetric analysis (TGA) curve of Chito and Chito-TZ.

**Table 2.** Thermal analysis and kinetic decomposition data of Chito-TZ in comparison with Chito.

| Treatment | Decomposition Stages | Decomposition Temp. Range | Wt. Loss (%) | [a] $T_{10}$ (°C) | [b] $T_{20}$ (°C) | $E_a$ (KJ mol$^{-1}$) | $R^2$ |
|---|---|---|---|---|---|---|---|
| Chito | 1 | 266.04–310.16 | 29.77 | 133.8 | 276.5 | 95.57 | 0.991 |
| | | 141.9–233.2 | 14.69 | | | 24.70 | 0.966 |
| Chito-TZ | 3 | 233.2–245.3 | 8.87 | 178.5 | 226.2 | 86.90 | 0.995 |
| | | 245.3 -332.6 | 16.00 | | | 18.62 | 0.963 |

[a] Temperature (°C) of 10% weight loss. [b] Temperature (°C) of 20% weight loss.

Broido's (BR) method is used to predict the thermal activation energy required to decompose chito and its derivative using the mathematical equation shown below:

$$\log[-\log(1-\alpha)] = -\frac{E_a}{2.303\ RT} + \log\left[\frac{ART_s^2}{\beta E_a}\right] \tag{4}$$

where $\alpha$ is the degree of sample decomposition at temperature $T$ (Kelvin) and is obtained using $(m_o - m_t)/(m_o - m_e)$, $A$ is the Arrhenius pre-exponential factor (min$^{-1}$), $R$ is the gas constant (8.314 Jmol$^{-1}$K$^{-1}$), $T_s$ is the temperature at the maximum degradation rate, $\beta$ is the heating rate (°C min$^{-1}$), $E_a$ is the activation energy of decomposition (KJ mol$^{-1}$), $m_o$ is the initial sample mass under decomposition, $m_t$ sample mass at a particular temperature, and $m_e$ is the sample mass at the end of decomposition step. The plot of $\log[-\log(1-\alpha)/T^2]$ versus $1000/T$ was applied to determine $E_a$ from the slope of the obtained straight lines (Figure 6). The obtained values along with correlation coefficients $R^2$ are presented in Table 2. As presented, lower activation energy values for Chito-TZ were observed compared with that of Chito; this proves that the thiazole moiety has a motivating effect on Chito-TZ decomposition.

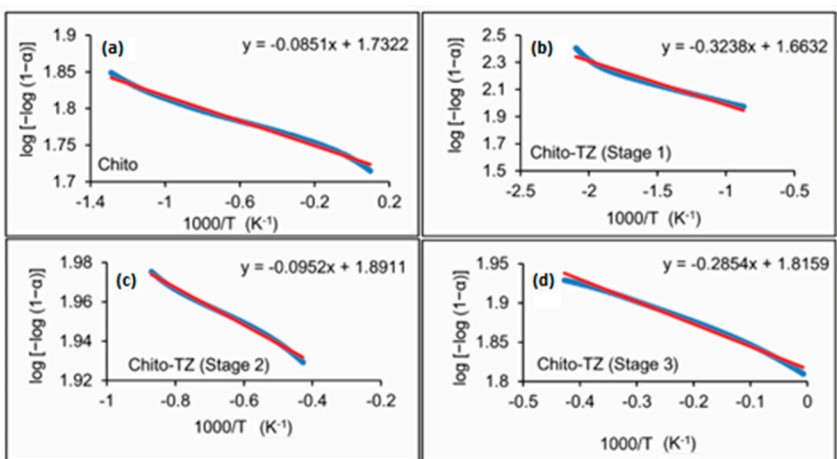

**Figure 6.** Broido's (BR) kinetic plots of decomposition for Chito (**a**), Chito-TZ stage 1 (**b**), Chito-TZ stage 2 (**c**), and Chito-TZ stage 3 (**d**). Blue curves—Broido's plots, red lines—the best fit lines]

*3.6. XRD Analysis*

Chitosan is a biopolymer with a highly stable structure because its chains are arranged in crystalline zones that coexist with amorphous zones when it is solid [2]. Here, Chito showed in Figure 7 XRD sharp peaks at $2\theta = 9.6°$ (020 plane) and $2\theta = 20°$ (110 plane), in line with the previous reports [38–40], which indicate that chitosan chains are highly ordered in a crystalline pattern. This ordered pattern of chitosan chains was interrupted after chemical modification with the thiadiazole derivative units. This fact is evidenced by the disappearance of crystalline peaks at $2\theta = 9.6°$ and $2\theta = 20°$ in the XRD pattern of Chito-TZ compared with Chito. The reason behind the reduced crystallinity of Chito-TZ is probably the interruption of the inter-/intra-molecular hydrogen bonds established via the $NH_2$/OH groups in Chito chains [41,42].

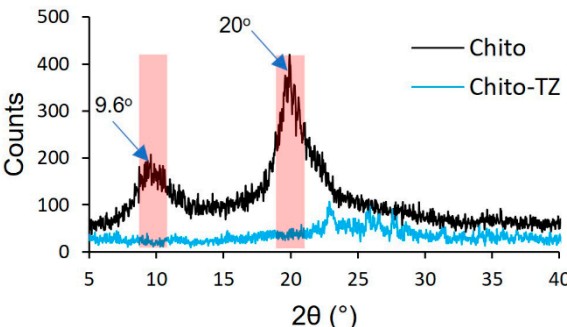

**Figure 7.** XRD analysis of Chito and Chito-TZ.

*3.7. XPS Analysis*

Figure 8 shows the XPS-wide scans of the Chito and Chito-TZ. The peaks of sulfur signals at around 229.40 eV (S 2s) and 165.09 eV (S 2p) in the spectrum of the Chito-TZ conjugate provide another evidence for the successful modification of chitosan with the sulfur-containing compound used in this study, 4-((5-(methylthio)-1,3,4-thiadiazol-2-yl)amino)-4-oxobutanoic acid. On the other hand, after conjugate formation, the peaks of oxygen (O *1s*), nitrogen (N *1s*), and carbon (C *1s*) in Chito were slightly shifted from 532.82 eV, 400.13 eV, and 286.99 eV to 533.29 eV, 401.21 eV, and 287.35 eV, respectively. The peak of S *2p* in the range 162–168 eV bending energy could be deconvoluted to two peaks at 163.9 and 165.4 eV (Figure 9) attributed to S-C (aliphatic) and S-C (thiadiazole), respectively. N1s spectra for chitosan showed two peaks appearing at binding energies of 399 and 400.5 eV, which are attributed to primary $NH_2$ and NH of acetamido group, respectively.

After the modification reaction, shifting to higher binding energies was observed at ~402 eV, which may correspond to the -N=C chitosan-thiadiazole conjugate (Figure 9).

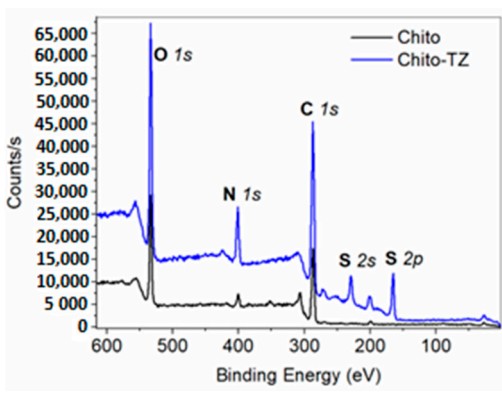

**Figure 8.** Wide XPS scans of Chito and Chito-TZ.

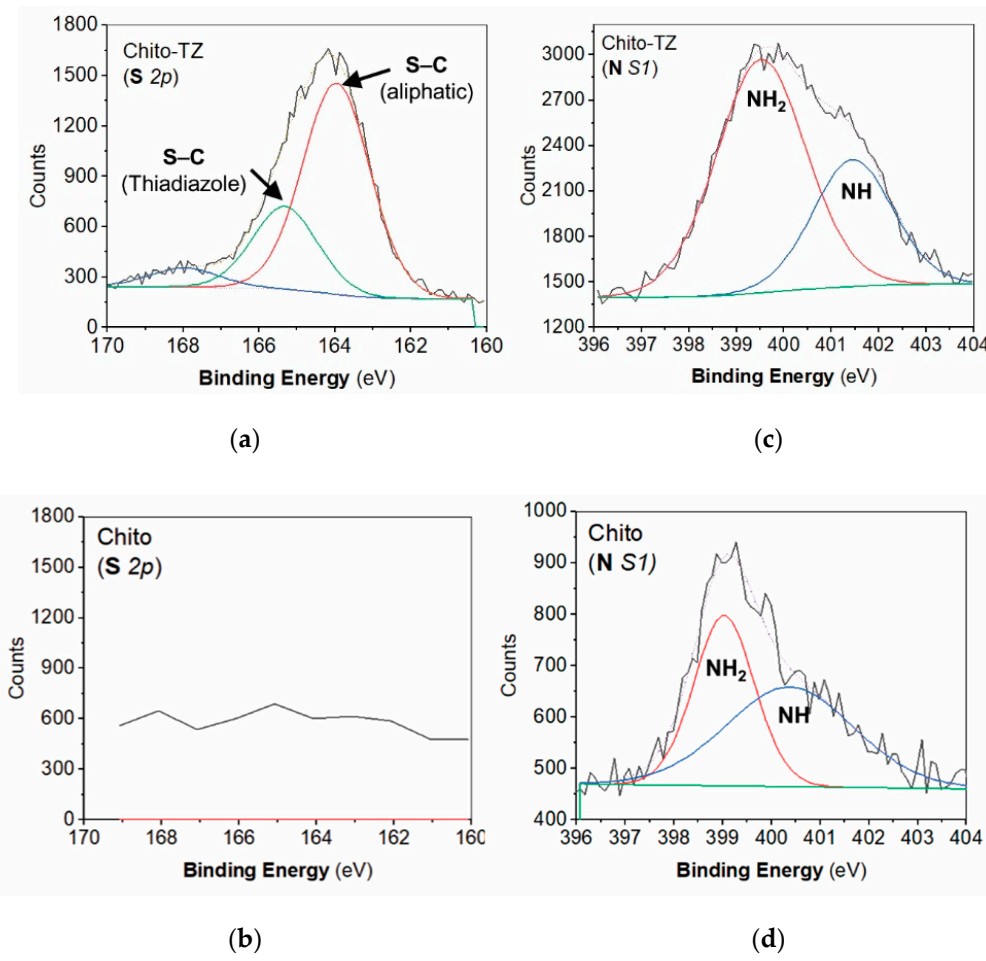

**Figure 9.** Sulfur (S 2p) for (**a**) Chito-TZ and (**b**) Chito; Nitrogen (N 1s) for (**c**) Chito-TZ and (**d**) Chito XPS spectra.

### 3.8. Biological Activity

3.8.1. Antimicrobial Activity

Discovering new antimicrobial active compounds is an important step for the enhancement of effective treatments for microbial infections. Finding novel chemicals with antimicrobial capabilities is more important than ever due to the emergence of antibiotic-resistant microorganisms. These active compounds can target and destroy harmful mi-

crobes, stopping the spread of infections and potentially saving lives [43]. Testing the efficacy of synthetic chemicals, natural products, and biologics against different pathogenic microbes is a key part of the quest for new antimicrobial drugs [44]. The main step in constructing new antibiotics is the discovery of novel compounds, which are crucial to the treatment of infections caused by antibiotic-resistant microbes [45].

Chitosan is a natural polysaccharide of chitin that exists in insect and crustacean exoskeletons. This polysaccharide has been characterized by its antimicrobial activity against a wide range of positively charged groups introduced to enhance its antimicrobial activity by increasing their bonding with the bacterial cell wall [46,47]. This functionalized or modified chitosan provides a new, effective, natural, and sustainable compound for combating microbial infections.

The obtained data suggest that chitosan (Chito) and thio-thidiazole-modified chitosan (Chito-TZ) have antimicrobial activity against pathogenic Gram-positive bacteria, Gram-negative bacteria, and unicellular fungi and that this activity is dependent on specific concentrations. This finding is compatible with published studies. For instance, the antimicrobial activity of chitosan derived from the exoskeleton of larvae, pupa, and dead adults of *Hermetia illucens* insects as well as commercial chitosan against *E. coli* and *Micrococcus flavus* was concentration-dependent [48]. At a concentration of 300 $\mu$g mL$^{-1}$, the results indicate that the non-modified Chito has high activity against *Bacillus subtilis* (with an inhibition zone of $16.7 \pm 0.6$ mm) compared to the modified one, which formed a clear inhibition zone of $15.3 \pm 0.5$ mm (Figure 10A), where there was no significant difference between the inhibition zones formed towards *Staphylococcus aureus* after treatment with this high concentration (300 $\mu$g mL$^{-1}$) of Chito and Chito-TZ (Figure 10B). Interestingly, the Gram-negative bacteria were more susceptible to Chito-TZ than Chito. As shown in Figure 10, the inhibition zones formed due to treatment with 300 $\mu$g mL$^{-1}$ of Chito and Chito-TZ were ($14.3 \pm 0.5$ and $17.7 \pm 0.6$ mm) and ($14.6 \pm 0.5$ and $16.7 \pm 0.6$ mm) for *Pseudomonas aeruginosa* and *E. coli*, respectively (Figure 10C,D).

Overall, the analysis of variance showed that at concentrations below 300 $\mu$g mL$^{-1}$, both Chito and Chito-TZ were more effective against Gram-negative bacteria compared to Gram-positive bacteria. This finding could be attributed to the fact that the cell walls of Gram-negative bacteria may be more susceptible to Chito and Chito-TZ at low concentrations. For instance, at a concentration of 100 $\mu$g mL$^{-1}$, the inhibition zones formed due to Chito and Chito-TZ treatment were in the ranges of 9.7–10.6 mm against Gram-positive bacteria, whereas it was in the ranges of 10.7–13.0 mm for Gram-negative bacteria. Interestingly, data analysis revealed that the activity of Chito at all tested concentrations was higher compared to Chito-TZ. For instance, the inhibition zones formed due to the treatment of *C. albicans* with varied concentrations (300–50 $\mu$g mL$^{-1}$) were in the ranges of 10.3–14.7 mm compared to those formed by Chito-TZ, which were 8.3–13.3 mm (Figure 10E). This suggests that the modified chitosan with the thio-thidiazole derivative may affect its activity against certain microorganisms.

The antibacterial activity of chitosan was varied and dependent on many factors, such as the molecular weight of chitosan, pH value, temperature, source of chitosan, type of organism, chitosan concentration, chemical modification (chitosan derivatives), and deacetylation degree (DD) [46,49]. There are numerous studies that investigated the antimicrobial activity of chitosan and its modification derivatives; some of them are compatible with the obtained finding, while others have reported different data. For instance, Devlieghere et al. showed that the sensitivity of Gram-negative bacteria towards chitosan was high compared to the sensitivity of Gram-positive bacteria and yeasts [50], whereas Fernandez-Saiz et al. reported that the sensitivity of *S. aureus* towards chitosan-based films was higher than the sensitivity of Gram-negative bacteria, *Salmonella* spp. [51]. However, chitosan has shown antimicrobial activity against Gram-negative and Gram-positive bacteria. This activity depends on the structure of microbial cell walls. Chitosan interacts with anionic structures that exist on the surface of Gram-negative bacteria, such as

proteins and lipopolysaccharides, whereas it interacts directly with cell wall layers of Gram-positive bacteria that bear negative charges, such as teichoic acids and peptidoglycans [52].

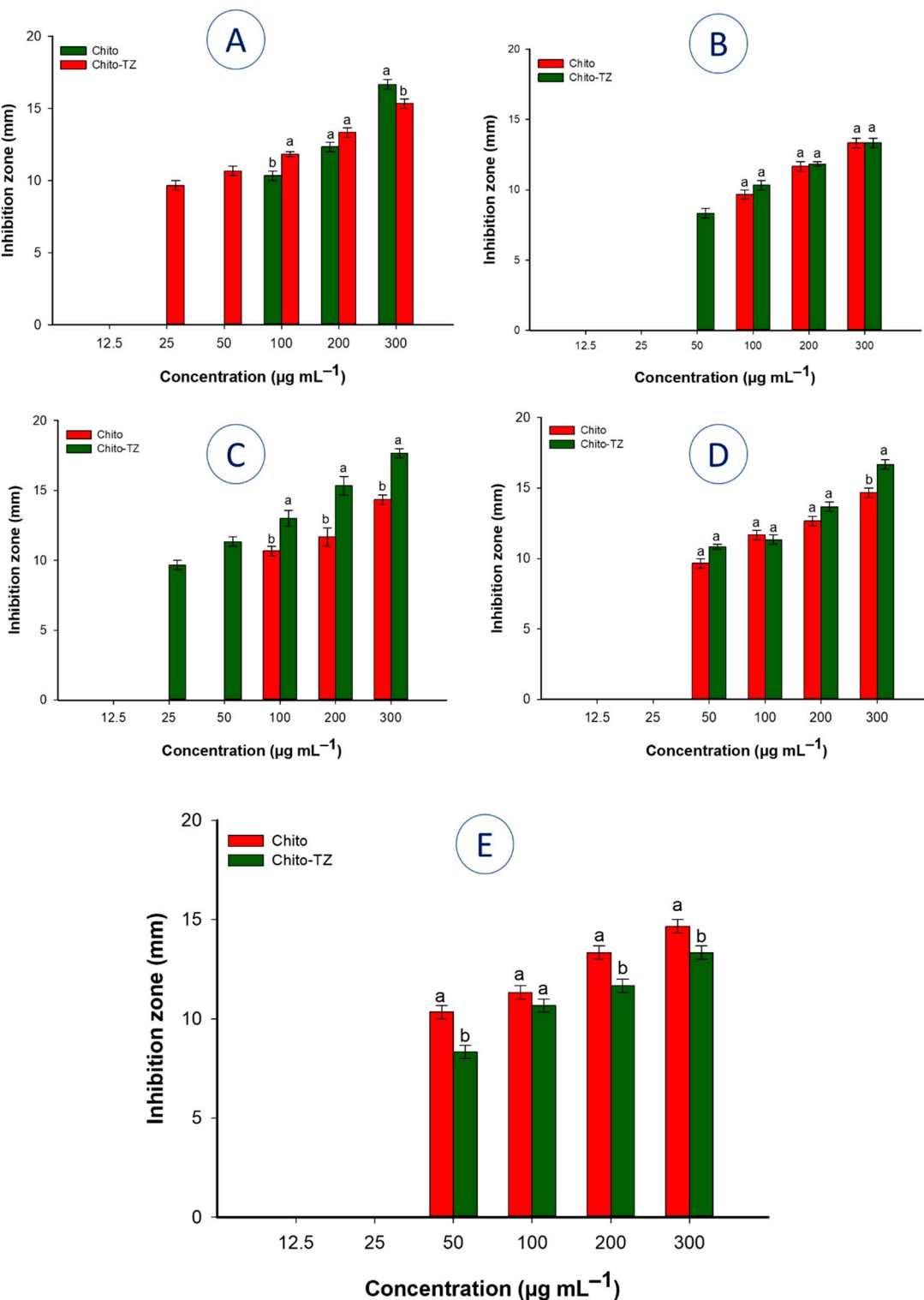

**Figure 10.** Antimicrobial activity of Chito and Chito-TZ against pathogenic Gram-positive bacteria, Gram-negative bacteria, and unicellular fungi. (**A**) is *B. subtilis*, (**B**) is *S. aureus*, (**C**) is *P. aeruginosa*, (**D**) is *E. coli*, and (**E**) is *C. albicans*. Different letters (a and b) on bars of Chito and Chito-TZ at the same concentration indicate that the mean values are significantly different ($p \leq 0.05$) ($n = 3$).

Regarding the antibacterial mechanisms of chitosan and its derivatives, there are several mechanisms, including the following: (A) The first mechanism is the electrostatic attraction between the cationic $NH_3^+$ present in chitosan and the negative charge of the bacterial cell membrane. This interaction changes the selective permeability of cytoplasmic membrane intra- and extracellular materials without any control [53,54]. This hypothesis was confirmed by Li and co-authors, who reported that the antibacterial activity of O-quaternary ammonium N-acyl thiourea chitosan (chitosan derivative) was attributed to the electrostatic attraction between the bacterial cell wall negative charge and the $NH_3^+$ group positive charge of chitosan derivative, which leads to increased membrane damage and selective permeability function [55]. (B) The second mechanism is the blocking of mRNA and protein synthesis due to the interaction of chitosan and its derivative residue with microbial DNA [53]. (C) Chitosan can destroy the membrane energy stability due to interaction with the cell membrane and disrupt the electron transport chain [56]. (D) Microbial growth can be inhibited due to the ability of chitosan to chelate nutrients and essential metal ions such as iron, copper, zinc, magnesium, and cadmium [57]. (E) Microbial growth can be inhibited via chitosan treatment due to its ability to form a polymer membrane on the cell surface, preventing the entrance of nutrients as well as inhibiting aerobic bacterial growth by acting as an oxygen barrier [52,58].

The treatment method selected for infection control relies on an accurate assessment of MIC [59]. Data analysis showed that the MIC value for Chito was 100 µg mL$^{-1}$ against *B. subtilis*, *S. aureus*, and *P. aeruginosa* with inhibition zones in the ranges of 9.7–10.6 mm, 50 µg mL$^{-1}$ for *E. coli* and *C. albicans* with inhibition zones of 9.7 ± 0.6 and 10.3 ± 0.6 mm, respectively. On the other hand, the MIC value for modified Chito-TZ was decreased to 25 µg mL$^{-1}$ for *B. subtilis* and *P. aeruginosa*, and 50 µg mL$^{-1}$ for *S. aureus*, *E. coli*, and *C. albicans* with varied clear zones (Figure 10). As shown in Figure 10, the modified chitosan (Chito-TZ) remains active against different pathogenic microbes at low concentrations based on MIC values compared to the non-modified one. Recently, the MIC values of chitosan varied in the ranges of 100 and 50 µg mL$^{-1}$ against Gram-positive bacteria (*B. subtilis* and *S. aureus*), Gram-negative bacteria (*P. aeruginosa* and *E. coli*), and unicellular fungi (*C. albicans*). These values were decreased in the case of a new thiadiazole chitosan derivative (BuTD-CH) to be in the ranges of 25–50 µg mL$^{-1}$ for the same organisms [60].

### 3.8.2. Biocontrol of Phytopathogens

Biodegradable and non-toxic chitosan biopolymer has shown promise as an alternative substance for synthetic pesticides in the management and control of phytopathogen-caused plant diseases [61]. Recently, the potential of chitosan to induce plant systematic resistance and hence inhibit the growth and control the development and spread of a wide range of plant pathogenic microbes has been investigated. Moreover, the modification of chitosan with different functional groups leads to expanding the incorporation of chitosan and its derivatives in the biocontrol of plant diseases caused by microbes, and its applications have been found to be more effective and eco-friendlier. In this current study, the efficacy of Chito and Chito-TZ to inhibit the growth of three phytopathogenic fungi designated as *Aspergillus niger*, *Fusarium oxysporum*, and *Fusarium solani* has been investigated (Figure 11A). The antifungal activity was achieved at concentrations of 100, 200, and 300 µg mL$^{-1}$ with concentration dependence matter (Figure 11B–D). This finding was compatible with those reported by Tan et al., who showed that the antifungal activity of chitosan and 1,2,3-triazolium-chitosan derivatives against three phytopathogenic fungi, *Colletotrichum lagenarium*, *Watermelon fusarium*, and *Fusarium oxysporum*, was attained in a concentration-dependent manner [13]. Analysis of variance showed that at high concentrations (300 µg mL$^{-1}$), the inhibition percentages between Chito and Chito-TZ were significantly different ($p \leq 0.05$). At this concentration, the inhibition percentages caused by Chito were 35.4 ± 0.7, 39.4 ± 1.7, and 37.6 ± 1.8% for *A. niger*, *F. oxysporum*, and *F. solani*, respectively. These inhibition percentages were increased in the presence of Chito-TZ to 45.2 ± 1.2, 47.7 ± 1.3, and 52.1 ± 1.3% for the same previous fungal strains. Similarly, Li

and coauthors succeeded in forming three thiadiazole chitosan derivatives, including 1,3,4-thiadiazole (TPCTS), 2-phenyl-1,3,4-thiadiazole (PTPCTS), and 2-methyl-1,3,4-thiadiazole (MTPCTS) and investigated their antifungal activity against three plant pathogenic fungi, namely *Colletotrichum lagenarium*, *Phomopsis asparagi*, and *Monilinia fructicola*, at various concentrations (01, 0.5, and 1.0 mg mL$^{-1}$) compared to chitosan without any modification [21]. The authors reported that the antifungal activity of chitosan and its thiadiazol derivatives was concentration dependent. The highest inhibition percentages were attained at 1.0 mg mL$^{-1}$ with values of 31.6, 31.3, and 9.8%; 58.6, 68.8, and 51.3%; 75.1, 77.6, and 60.6%; and 75.3, 82.5, and 65.8% for chitosan, TPCTS, PTPCTS, and MTPCTS, respectively, for the pathogenic fungi *C. lagenarium*, *P. asparagi*, and *M. fructicola*, respectively.

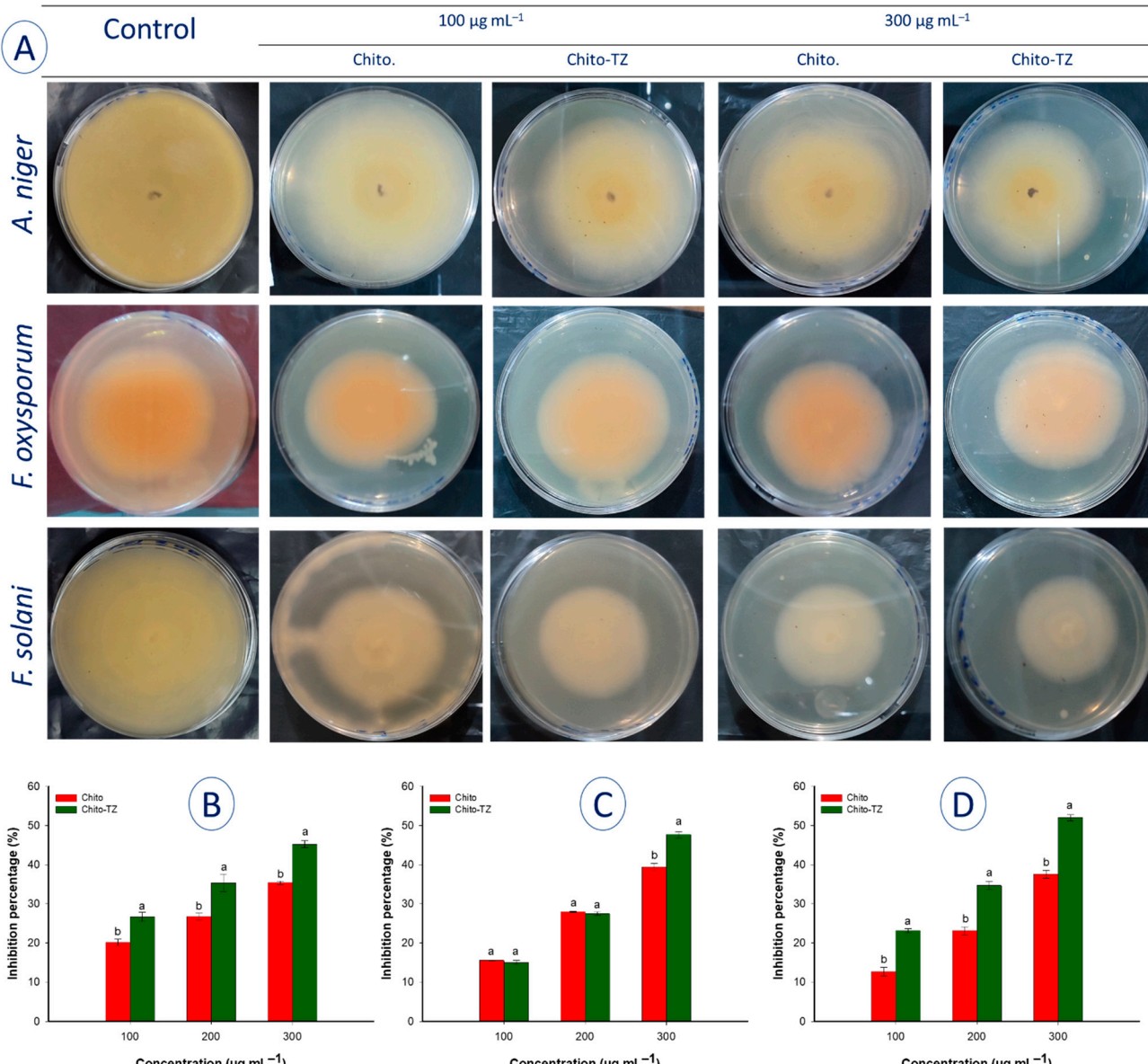

**Figure 11.** Antifungal activity of Chito and Chito-TZ towards phytopathogenic strains. (**A**) is the radial growth of phytopathogenic fungi grown on PDA supplemented with different concentrations of Chito and Chito-TZ compared to control, (**B**–**D**), are antifungal activity against *A. niger*, *F. oxysporum*, and *F. solani*, respectively. Different letters (a and b) on bars of Chito and Chito-TZ at the same concentration indicates that the mean values are significantly different ($p \leq 0.05$) ($n = 3$).

At low concentrations (100 and 200 µg mL$^{-1}$), the antifungal activity caused by Chito-TZ was significantly different ($p \leq 0.001$) against *A. niger* and *F. solani* compared to Chito, whereas the activity of Chito and Chito-TZ against *F. oxysporum* was not significant ($p = 0.099$). As shown in Figure 11, Chito inhibits the growth of *A. niger* and *F. solani* with percentages of $26.7 \pm 1.5$ and $23.1 \pm 1.8\%$ and $20.1 \pm 1.5$ and $12.1 \pm 1.7\%$, respectively, at concentrations of 200 and 100 µg mL$^{-1}$. These inhibition percentages showed a significant increase in the presence of Chito-TZ to be $35.3 \pm 3.9$ and $34.7 \pm 1.7\%$ and $26.7 \pm 2.0$ and $23.1 \pm 0.9\%$ for the same previous organisms and concentrations, respectively. On the other hand, the growth inhibition of *F. oxysporum* with chitosan was $27.5 \pm 0.3\%$ and $15.2 \pm 0.1\%$ for concentrations of 200 and 100 µg mL$^{-1}$, respectively. These values show no significant difference in the presence of chitosan derivative (Chito-TZ) at $27.7 \pm 0.8\%$ and $15.5 \pm 1.1\%$ for concentrations of 200 and 100 µg mL$^{-1}$, respectively. The obtained results confirmed that the modified chitosan with thio-thiadiazole derivative may affect the growth of some plant pathogenic fungi in higher percentages than non-modified chitosan. This phenomenon could be attributed to the presence of thiadiazole functional groups. Some authors reported that the activity of chitosan derivatives to inhibit the growth of plant pathogenic fungi increased based on the length of the alkyl substituent [21,62].

It has been shown that chitosan and its thiadiazole derivatives have diverse mechanisms to inhibit the growth of plant pathogenic fungi, including enhancing plant defense, destroying the fungal cell wall, and inhibiting fungal enzymatic production. Phytoalexins and reactive oxygen species (ROS) are two examples of plant defense mechanisms that can be triggered by chitosan and its derivatives. Phytoalexins are plant-made chemicals that reduce fungal development when exposed to infection. Also, ROS causes oxidative damage and hence cell death [63]. Because of their positive charge, chitosan and its thiadiazole derivatives can destroy the fungal cell wall by interacting with negatively charged components like glucan and chitin, resulting in ultimate cell death because intracellular components leak out [64]. The antifungal activity of chitosan and its derivatives is proportional to DD; the activity increases with higher DD [65]. In this current study, due to the high DD of the used chitosan, we predict its high antifungal activity. The inhibition of fungal enzymatic activity is an important antifungal mechanism for thiadiazole-chitosan derivatives. For instance, these derivatives have an inhibitory effect on chitin synthase, which has a critical role in the synthesis of the fungal cell wall [21].

3.8.3. Seed Germination

The toxicity of Chito and Chito-TZ conjugate on the seedling and morphological characteristics was evaluated using broad beans. As shown, chitosan and its derivative have a positive effect on the morphological characteristics, with improvements in germination rate, seedling length, and fresh and dry weight of shoot and root for chitosan compared with thiadiazole-derivative and positive control (tap water). Data analysis showed that the maximum improvement in the length and weight of broad bean shoot and root was attained at a concentration of 200 µg mL$^{-1}$ of Chito and Chito-TZ (Table 3). Similarly, the seedling length and fresh and dry weight of cucumber were higher at a concentration of 0.5 mg mL$^{-1}$ of chitosan and its derivatives compared to control (tap water) and concentrations of 0.1 and 1.0 mg mL$^{-1}$ [13]. Also, the maximum germination percentages and higher SVI were obtained at the same concentration (200 µg mL$^{-1}$).

Hormonal regulation, improved water uptake, enzymatic activities, and antioxidant activity are the main mechanisms utilized in chitosan and its derivatives to enhance seed germination. Plant hormones such as gibberellic acid (GA) and abscisic acid (ABA) can be influenced by chitosan and its derivatives. The germination percentages were increased by GA by breaking the dormancy statement and stimulating the growth of the embryo, whereas the ABA inhibits the seed germination process. It was shown that the treatment with chitosan and its thiadiazole derivatives has the efficacy to increase the level of GA and decrease the ABA production, thereby improving seed germination [66]. By increasing the seed coat's permeability to water, chitosan and its derivatives facilitate more rapid and

complete water uptake of the seed. This increases moisture and initiates metabolic processes that improve seed germination [67]. The activity of certain enzymes, such as protease and alpha amylase, can be enhanced by treatment with chitosan and its derivatives. The amino acids necessary for protein synthesis are provided via the breakdown of storage proteins by the action of protease enzymes, whereas the sugars needed to supply the cells with energy for seedling growth can be obtained by degrading starch with amylase enzymes. The expression and activities of these enzymes can be positively influenced by chitosan and its derivatives [68]. Interestingly, due to the antioxidant activity of chitosan and its derivatives, it can scavenge the ROS and decrease lipid peroxidation, thereby protecting the seed during germination from oxidative damage [69]. It is worth noting that the precise mechanisms at play may change based on the seed types, the concentration of chitosan, and the derivatives used.

**Table 3.** Effect of chitosan and its thiadiazole derivatives in seed germination of broad bean.

| Concentration | Treatment | Germination Percentages (%) | Shoot Length (cm) | Root Length (cm) | Shoot Weight (mg) | Root Weight (mg) | SVI * |
|---|---|---|---|---|---|---|---|
| Control | $H_2O$ | $87.0 \pm 0.7$ | $11.7 \pm 0.8$ | $11.1 \pm 1.5$ | $1080.0 \pm 5.6$ | $711.0 \pm 6.6$ | 1018 |
| 100 µg mL$^{-1}$ | Chito | $90.0 \pm 0.5$ | $15.3 \pm 1.5$ | $12.7 \pm 11.2$ | $1186.7 \pm 22.7$ | $761.0 \pm 16.1$ | 1377 |
| | Chito-TZ | $92.0 \pm 0.6$ | $13.3 \pm 0.6$ | $10.9 \pm 0.6$ | $1104.0 \pm 7.6$ | $726.0 \pm 4.6$ | 1224 |
| 200 µg mL$^{-1}$ | Chito | $96.0 \pm 1.1$ | $17.2 \pm 0.3$ | $15.0 \pm 1.0$ | $2337.7 \pm 26.1$ | $997.0 \pm 22.3$ | 1651 |
| | Chito-TZ | $94.0 \pm 0.9$ | $14.8 \pm 0.3$ | $12.7 \pm 0.7$ | $1362.7 \pm 9.5$ | $814.7 \pm 18.8$ | 1391 |
| 300 µg mL$^{-1}$ | Chito | $90.0 \pm 0.5$ | $14.0 \pm 1.3$ | $14.7 \pm 0.6$ | $1807.7 \pm 16.4$ | $980.7 \pm 20.8$ | 1260 |
| | Chito-TZ | $90.0 \pm 0.5$ | $13.3 \pm 0.6$ | $11.2 \pm 1.5$ | $1215.3 \pm 22.5$ | $733.3 \pm 20.1$ | 1197 |

* SVI is seed vigor index.

## 4. Conclusions

Based on the above results of IR, NMR, SEM, TGA, XRD, and XPS, we can conclude the successful synthesis of a new chitosan-thio-thidiazole conjugate (Chito-TZ). The biological activities of Chito and Chito-TZ, including antimicrobial activity, antifungal against phytopathogenic fungi, and seed germination, were investigated and exhibited these activities in a concentration-dependent manner. The newly functionalized Chito-TZ showed promising antibacterial activity toward Gram-positive bacteria, Gram-negative bacteria, and unicellular fungi compared with non-modified chitosan. Also, the inhibition percentages of three phytopathogenic fungi designated as *Aspergillus niger*, *Fusarium oxysporum*, and *Fusarium solani* were assessed at different concentrations (100, 200, and 300 µg mL$^{-1}$). Data analysis showed that the maximum inhibition was attained at a concentration of 300 µg mL$^{-1}$ with the percentages $35.8 \pm 0.7$, $39.4 \pm 1.7$, and $37.6 \pm 1.8\%$ and $45.2 \pm 1.6$, $47.7 \pm 1.3$, and $52.1 \pm 1.1\%$ for *A. niger*, *F. oxysporum*, and *F. solani*, respectively. Interestingly, the highest germination percentages, shoot and root length, shoot and root weight, and SVI of broad beans were achieved due to treatment with Chito at a concentration of 200 µg mL$^{-1}$ compared to modified chitosan. Overall, the functionalized chitosan with thio-thidiazole improved its activity against human pathogenic microbes and phytopathogenic fungi and can be utilized in biomedical and agricultural applications.

**Author Contributions:** Conceptualization, A.G.I., W.E.E., S.M.H. and A.F.; methodology, A.G.I., W.E.E., A.M.E., A.E.M. and A.F.; software, A.G.I., W.E.E., A.M.E., A.E.M., S.M.H. and A.F.; validation, A.G.I., W.E.E., M.A., A.A.M.A. and A.F.; formal analysis, A.G.I., W.E.E., A.M.E. and A.F.; investigation, A.G.I., W.E.E., S.M.H. and A.F.; resources, W.E.E., A.E.M., M.A. and A.A.M.A.; data curation, A.G.I., S.M.H. and A.F.; writing—original draft preparation, A.G.I., W.E.E. and A.E.M.; writing—review and editing, A.G.I. and A.F.; visualization, A.G.I., W.E.E., A.M.E., M.A., A.E.M., A.A.M.A., S.M.H. and A.F.; supervision, A.G.I., W.E.E., S.M.H. and A.F.; project administration, A.G.I., S.M.H. and A.F.; funding acquisition, M.A. and A.A.M.A. All authors have read and agreed to the published version of the manuscript.

**Funding:** This research received no external funding.

**Institutional Review Board Statement:** Not applicable.

**Informed Consent Statement:** Not applicable.

**Data Availability Statement:** The data presented in this study are available upon request from the corresponding author.

**Acknowledgments:** The authors extend their appreciation to the Faculty of Science, Al-Azhar University, Cairo, Egypt, for the great support to achieve and publication of this research work. Also, the authors thank the Princess Nourah bint Abdulrahman University Researchers Supporting Project number (PNURSP2023R182), Princess Nourah bint Abdulrahman University, Riyadh, Saudi Arabia.

**Conflicts of Interest:** The authors declare no conflict of interest.

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
