# Peer review of "New Functionalized Chitosan with Thio-Thiadiazole Derivative with Enhanced Inhibition of Pathogenic Bacteria, Plant Threatening Fungi, and Improvement of Seed Germination"

_chemistry, doi:10.3390/chemistry5030118_

Round 1

Reviewer 1 Report

In this paper, the authors describe the synthesis and characterization of new chitosan-thio-thiadizole conjugate as well as an interesting study of the biological activities of this conjugate compared to the native chitosan.

Although there are many papers in the literature on chitosan and its derivatives, the present paper is well written and it might be interesting and useful for researchers in the antimicrobial compounds field. The material presented is selected, synthesized and organized in such a way that it is easy to follow, giving a clear image of the purpose and results achieved. Also, the relevant conclusions were drawn. I recommend the paper publication in present form.

Author Response

Dear reviewer, thank you for your valuable comments. The manuscript was revised according to reviewers' comments point-by-point. 

Reviewer 2 Report

1 The methods for modifying chitosan mentioned in the abstract do not mention that they should be briefly described.

2 2.2.2“the appropriately saturated methyl iodide was added for a period of 15 minutes”How to judge the saturation of methyl iodide is needed in this paper? Second, does the saturation of methyl iodide affect the product?

3 2.4.2 Formula (1) Unified format, the formula should be "×100%".

4 3.2 Figure 13.4 Figure 4 Not highlight the pointFigure 4 SEMThe information bar below the image should hide the re-add scale.

5 3.5 TGA used to confirm that the product is Chito-TZ? If so, it can be confirmed that the product is different from Chito, but how do you determine by TGA that the product is Chito-TZ?

6 3.5 The formula for calculating α should also be typeset in the text according to the formula format.

7 3.8.1 Figure 10What is the meaning of a and b in the figure? In addition, several antibacterial mechanisms of chitosan are described in this part, but what is the antibacterial mechanism of the material prepared in this paper?

8 There is no need for a "p." before the page number in the bibliographic format; In addition, the reference format is unified, such as: journal author. Title [J]. Publication title, publication year, volume (issue): Starting and ending page numbers.

Some English expressions in the text needs improvement

Author Response

(The authors gave the same response as above.)

Reviewer 3 Report

Dear authors,

I revised your manuscript about synthesizing a new thiadiazole derivative of a compound that has been continuously worked on by your research group. 

This research has important outcomes, but your published research also promised that chitosan-derived compounds with thiadiazole will be yielded more antimicrobial activities. Therefore, the results are expected.

On the other hand, the study is worth to be published after minor revisions since it has partial novelty and cited potential.

You find my comments below:

Line 118. Please write the open form of RT for the first time.

Line 196. Please use the multiplication symbol.

Please add some discussions or literature citations in the 3.2 Spectral analysis FTIR section.

The value between T10 and T20 for Chito is 142.7°C while it is 47.7°C for Chito-TZ. Could you please add some discussions to the manuscript related to those differences and structure stability?

Please check Figure 5 A legend. Is the color red for Chito or Chito-TZ compound?

In the seed germination section, Table 3 shows that the control section has both the results belonging to Chito and Chito-TZ. However, the seeds of control are grown only in sterilized dH2O. Could you please clarify this situation?

Author Response

(The authors gave the same response as above.)
